# High-Risk Obesity Phenotypes: Target for Multimorbidity Prevention at the ROFEMI Study

**DOI:** 10.3390/jcm11164644

**Published:** 2022-08-09

**Authors:** Juana Carretero-Gómez, Pablo Pérez-Martínez, José Miguel Seguí-Ripoll, Francisco Javier Carrasco-Sánchez, Nagore Lois Martínez, Esther Fernández Pérez, Onán Pérez Hernández, Miguel Ángel García Ordoñez, Candelaria Martín González, Juan Francisco Vigueras-Pérez, Francesc Puchades, María Cristina Blasco Avaria, María Isabel Pérez Soto, Javier Ena, José Carlos Arévalo-Lorido

**Affiliations:** 1Internal Medicine Department, University Hospital of Badajoz, 06085 Badajoz, Spain; 2Internal Medicine Department, IMIBIC/Reina Sofia University Hospital, University of Cordoba, 14004 Cordoba, Spain; 3CIBER Fisiopatología Obesidad y Nutrición (CIBEROBN), Instituto de Salud Carlos III, 28029 Madrid, Spain; 4Internal Medicine Department, San Juan de Alicante University Hospital, 03550 San Juan de Alicante, Spain; 5Internal Medicine Department, Juan Ramón Jiménez University Hospital, 21005 Huelva, Spain; 6Internal Medicine Department, Severo Ochoa University Hospital, 28911 Leganés, Spain; 7Internal Medicine Department, León University Hospital, 24071 León, Spain; 8Internal Medicine Department, Nuestra Señora de la Candelaria University Hospital, 38010 Santa Cruz de Tenerife, Spain; 9Internal Medicine Department, Antequera University Hospital, 29200 Antequera, Spain; 10Internal Medicine Department, Canarias University Hospital, 38320 Santa Cruz de Tenerife, Spain; 11Internal Medicine Department, Vithas Hospital, 35005 Las Palmas de Gran Canarias, Spain; 12Internal Medicine Department, General Hospital of Valencia, 46014 Valencia, Spain; 13Internal Medicine Department, University Hospital of Manises, 46940 Manises, Spain; 14Internal Medicine Department, Vinalopó University Hospital, 03293 Elche, Spain; 15Internal Medicine Department, La Vila Joyosa University Hospital, 03570 Villajoyosa, Spain

**Keywords:** obesity, inflammation, adiposity, waist circumference, phenotypes

## Abstract

***Background***: Describe the profile of patients with obesity in internal medicine to determine the role of adiposity and related inflammation on the metabolic risk profile and, identify various “high-risk obesity” phenotypes by means of a cluster analysis. This study aimed to identify different profiles of patients with high-risk obesity based on a cluster analysis. ***Methods:*** Cross-sectional, multicenter project that included outpatients attended to in internal medicine. A total of 536 patients were studied. The mean age was 62 years, 51% were women. Patients were recruited from internal medicine departments over two weeks in November and December 2021 and classified into four risk groups according to body mass index (BMI) and waist circumference (WC). High-risk obesity was defined as BMI > 35 Kg/m^2^ or BMI 30–34.9 Kg/m^2^ and a high WC (>102 cm for men and >88 cm for women). Hierarchical and partitioning clustering approaches were performed to identify profiles. ***Results:*** A total of 462 (86%) subjects were classified into the high-risk obesity group. After excluding 19 patients missing critical data, two profiles emerged: cluster 1 (*n* = 396) and cluster 2 (*n* = 47). Compared to cluster 1, cluster 2 had a worse profile, characterized by older age (77 ± 16 vs. 61 ± 21 years, *p* < 0.01), a Charlson Comorbidity Index > 3 (53% vs. 5%, *p* < 0.001), depression (36% vs. 19%, *p* = 0.008), severe disability (64% vs. 3%, *p* < 0.001), and a sarcopenia score ≥ 4 (79% vs. 16%, *p* < 0.01). In addition, cluster 2 had greater inflammation than cluster 1 (hsCRP: 5.8 ± 4.1 vs. 2.1 ± 4.5 mg/dL, *p* = 0.008). ***Conclusions:*** Two profiles of subjects with high-risk obesity were identified. Based on that, older subjects with obesity require measures that target sarcopenia, disability, psychological health, and significant comorbidities to prevent further health deterioration. Longitudinal studies should be performed to identify potential risk factors of subjects who progress from cluster 1 to cluster 2.

## 1. Answer the Study Importance Questions

Body mass index and waist circumference do not provide enough information to assess patients with obesity for prevention or therapy needs.

In real world practice, obesity should be the target of management of multimorbidity.

Stratifying obesity into risk-based classifications by cluster allows us to identify individuals with “*high-risk obesity*”.

We found a cluster of elderly patients with obesity with a worse profile in general, psychological, and functional health than younger patients.

Longitudinal studies can identify risk factors in young patients with obesity to prevent further health decline.

## 2. Introduction

The WHO estimates that more than 650 million people have obesity worldwide [1]. Obesity predisposes individuals to a wide array of clinical conditions, including type 2 diabetes mellitus (T2DM), cardiovascular disease (CVD), chronic kidney disease (CKD), cancers, mental health disorders, and musculoskeletal disorders [2,3]. However, it is unclear whether these diseases are distributed among all people with obesity or are clustered in smaller groups of individuals with obesity-related multimorbidity [4]. Despite reports of obesity-related disease clustering, few studies have examined the role of obesity in the development of complex multimorbidity (defined as four or more comorbid diseases) and those that have been conducted mainly examine cardiometabolic comorbidities [5]. Previous cohort studies have found that the most prevalent disease clusters in individuals with obesity include joint disorders, dyslipidemia, T2DM, sleep disorders, and CKD. Excess adiposity is estimated to lead to the loss of three to ten years of life and entails an increased risk of premature death due to its role in the development of comorbidities, both cardiometabolic and others, as well as severe functional limitations that affect quality of life [6]. In light of these ramifications, stratifying obesity into risk-based categories would allow clinicians to better and earlier identify individuals with “high-risk obesity,” or those whose excess adiposity entails an increased risk to their health.

To date, most clinical and epidemiological studies have used body mass index (BMI) to classify obesity. At a minimum, this instrument must be adjusted for age, sex, and demographic background. Furthermore, as metabolic health risks are at least as much due to the overall fat mass as to its distribution in the body, it is particularly important to think beyond BMI and incorporate other tools for measuring the distribution of fat mass, such as waist circumference (WC), to better understand its role in obesity-related comorbidities. This is especially relevant after the fifth and sixth decade of life, when there is a tendency toward visceral and abdominal fat deposits [7,8]. In an extensive review, Visscher concluded that most evidence suggests a trend in which relative increases in WC were greater over time than relative increases in BMI in a manner seemingly independent of age, sex, or ethnicity [9]. Resistance to the inclusion of WC in routine clinical practice not only goes against evidence of its utility but may also lead to missing an opportunity to counsel patients regarding higher-risk obesity phenotypes. In addition, the measurement of both BMI and WC could provide unique opportunities to monitor the utility of treatment and effectiveness of interventions designed to manage obesity and related metabolic diseases.

The aim of this study was to describe the profile of patients with obesity in internal medicine departments in order to determine the role of adiposity and related inflammation on the metabolic risk profile and, based on this, identify various “high-risk obesity” phenotypes by means of a cluster analysis.

## 3. Materials and Methods

The ROFEMI (*Registro de Obesidad y sus Fenotipos en Medicina Interna*, Registry of Obesity and its Phenotypes in Internal Medicine) study is a cross-sectional, multicenter project that included outpatients attended to in internal medicine clinics in 46 Spanish hospitals. To avoid bias, hospitals started collecting data in one of the two selected date ranges (November 15 to 19, 29 November 2021 to 3 December 2021). All outpatients were given conventional treatment and medical care and were enrolled in the registry if they met the inclusion criteria, which were: patients 18 years of age or older, patients able to be weighed and measured, patients with either a BMI greater than 25 kg/m^2^ or a WC greater than 94 cm for men or 80 cm for women, patients who gave their informed consent.

### 3.1. Patient Classification and Data Collection

Patients were recruited through the ROFEMI registry (https://rofemi.reginus.es) (accessed on 17 January 2022), which is sponsored by the Diabetes and Obesity Working Group of the Spanish Society of Internal Medicine (SEMI, for its initials in Spanish) and has been approved by the Ethics Committee.

Data were retrospectively collected and included sociodemographic, anthropometric, clinical (SARC-F sarcopenia score [10], Charlson Comorbidity Index [11], and Edmonton Obesity Staging System score [12], and laboratory variables grouped under various headings) data. The triglyceride-glucose index (TyG index) [13] was calculated as a measure of insulin resistance using the following formula: Ln (TG [mg/dL] × glucose [mg/dL]/2). Furthermore, the lymphocyte to C-reactive protein ratio (LCR) and the C-reactive protein to albumin ratio (CAR) were calculated as markers of inflammation [14,15].

Patients were classified by BMI and WC measurements according to the NICE guidelines (2021) [16] into the following categories (Figure 1, Flowchart):BMI: Healthy weight—BMI 18.5 kg/m^2^ to 24.9 kg/m^2^; Overweight—BMI 25 kg/m^2^ to 29.9 kg/m^2^; Obese—BMI 30 kg/m^2^ to 34.9 kg/m^2^; Very obese—BMI 35 kg/m^2^ or higher.WC: For men, low risk—less than 94 cm; high risk—94–102 cm; very high risk—greater than 102 cm. For women, low risk—less than 80 cm; high risk—80–88 cm; very high risk—greater than 88 cm.

With these two variables, patients were classified into four groups according to health risk: no increased risk (group 1), increased risk (group 2), high risk (group 3), and very high risk (group 4) (Figure 2).

### 3.2. Statistical Analysis

WC data was added for patients who were missing data in order to classify them (121, 22.6%). The k-nearest neighbors (KNN) algorithm (k-NN, k = 10), a non-parametric classification method that is sensitive to the local structure of the data, was used [17].

Qualitative variables were expressed as absolute numbers (percentage) and the chi-square test was used to compare them among groups. Quantitative variables were non-parametric and expressed as medians (interquartile range, IQR). In order to compare these variables among groups, the Kruskal-Wallis test was used for variables with similar variance (Levene’s test) and the ANOVA and Welch’s t-test were used for variables with unequal variance.

After the initial classification, we performed a second analysis specifically on the patients in group 4 (very high risk) in order to analyze potential differences among them in regard to the following variables: sex; education level; Edmonton Obesity Staging System score; Charlson Comorbidity Index; presence/absence of disability for basic activities of daily living, sarcopenia, hypertension, T2DM, hyperuricemia, heart failure, coronary artery disease, stroke, gastroesophageal reflux disease, obstructive lung diseases (chronic obstructive pulmonary disease or asthma), cancer, arthrosis, and depression. With a Hopkins statistic (measure of the clustering tendency of the data set) of 0.39, we grew an agglomerative hierarchical clustering with average linking to group the variables and identify aggregated conditions. The hclust function in R with the dissimilarity matrix defined by the binary distance (Figure 3) was used. Dichotomous variables (sex, sarcopenia, hypertension, T2DM, dyslipidemia, hyperuricemia, heart failure, coronary artery disease, stroke, gastroesophageal reflux disease, obstructive lung diseases, cancer, arthrosis, depression, disability, and educational level) were assigned a value of one when the variable was present and zero when absent. In regard to the Charlson Comorbidity Index, the severity of comorbidity was classified into three categories based on the score: mild for CCI scores of zero or one, which was assigned a value of one; moderate for a CCI score of two, which was assigned a value of two; and severe for CCI scores ≥ 3, which was assigned a value of three. Lastly, the Edmonton Obesity Staging System score was assigned its respective numerical categories as values. The optimal partition of the data was found so that all the constraints were satisfied. The optimal number of clusters (Figure 3) was two subsets.

Once the clusters were built, univariate comparisons were again performed among them. Qualitative variables of the cluster groups were expressed as absolute numbers and percentages and were compared using Fisher’s exact test or the chi-square test, as appropriate. Quantitative variables of the cluster groups were expressed as median and interquartile ranges and were compared using the Mann-Whitney U test.

All analyses were conducted using R version 3.3.2 (R Core Team 2020. R: A language and environment for statistical computing. R Foundation for Statistical Computing, Vienna, Austria. https://www.R-project.org/ (accessed on 17 January 2022)). Statistical significance was defined as *p* < 0.05.

## 4. Results

A total of 543 patients were included, of which 536 were analyzed (Figure 1). The median age (IQR) of the sample was 62 (22) years and there were slightly more women (51.4%) than men. The median (IQR) BMI was 34.1 (6.4) Kg/m^2^ and the median (IQR) WC was 110 (18) cm. The majority of patients had mild comorbidity (median 1, IQR 3), 28.7% had a disability, and 23.9% had sarcopenia.

### 4.1. BMI-WC Classification (Nice Guidelines)

According to the BMI-WC classification, three patients were at no increased risk (group 1), 27 patients were at increased risk (group 2), 44 patients were at high risk (group 3), and 462 patients were at very high risk (group 4). The distribution of the total sample and the distribution classified by sex is shown in Figure 2.

Group 4—the highest risk group—had a significantly greater presence of hypertension (*p* = 0.0008), T2DM (*p* = 0.03), dyslipidemia (*p* = 0.007), and hyperuricemia (*p* = 0.001). A significantly greater proportion of patients in this group received treatment with statins and anti-hypertensive drugs (*p* = 0.02 and *p* = 0.007, respectively) than groups 1, 2, and 3. There was a significantly higher proportion of patients with cancer in group 1 (*p* = 0.004), probably by chance (one patient from group 3 has cancer). Comorbidity, and consequently the number of drugs indicated on the treatment schedule, were also higher in groups 3 and 4 (*p* = 0.009 and *p* = 0.004, respectively).

With regard to laboratory findings, group 4 had significantly higher fasting glucose (*p* = 0.04), TyG index (*p* = 0.008), serum uric acid (*p* < 0.00), and triglyceride (*p* = 0.009) levels, but levels of inflammatory biomarkers such as hsCRP (*p* = 0.57), LCR (*p* = 0.78), or CAR (*p* = 0.78) were not higher in a statistically significant manner. Data on laboratory findings are shown in Table 1 and Table 2.

As Appendix A, the grouped analysis of groups 1, 2, and 3 versus group 4 is attached (Group 1 and 2). This sub-analysis, which does not follow the Nice guidelines, shows the benefit of classifying patients according to the different WC values (Appendix A).

### 4.2. Clusters of Very High-Risk Patients

Since most patients in the ROFEMI registry belonged to group 4, a cluster analysis was performed on this group alone to further explore potential differences among the patients in this group. A total of 446 patients who had data available on all the pre-specified variables were analyzed to grow the clusters (Figure 1).

Two clusters were obtained. Cluster 1 had 396 patients and cluster 2 had 47. Cluster 1 has previously been described (high-risk patient). Compared to cluster 1, patients in cluster 2 were older (median 77 years, IQR 16, *p* < 0.00) and had a higher WC (median 116.3 cm, IQR 15.7, *p* = 0.02). A higher percentage had CVD, such as coronary artery disease (*p* = 0.0006), stroke (*p* < 0.00), or heart failure (*p* = 0.0006). Likewise, they had more sarcopenia (*p* < 0.00) and disability (*p* < 0.00) (Graphical Abstract). With regard to laboratory findings, interestingly, the patients in cluster 2 had a greater inflammatory burden, with higher levels of CRP (*p* = 0.008), leukocytes (*p* = 0.007), CAR (*p* = 0.01), and LCR (*p* = 0.002). The rest of the findings are shown in Table 3.

## 5. Discussion

This work found that according to the BMI-WC classification, 86.22% of the sample (462 patients) belonged to the high-risk obesity group (group 4) (Figure 1 and Figure 2). The median BMI in group 4 was 34.1 (6.4) Kg/m^2^ (class 1 obesity) and the median WC was 110 (18) cm. This group had a significantly greater presence of hypertension (*p* = 0.0008), T2DM (*p* = 0.03), dyslipidemia (*p* = 0.007), and hyperuricemia (*p* = 0.001), but not established CVD (heart failure, coronary artery disease, or stroke) (Table 1). Accordingly, higher risk patients had higher levels of metabolic intermediates, such as fasting glucose (*p* = 0.04), TyG index (*p* = 0.008), and triglycerides (*p* = 0.009), but did not have higher levels of inflammatory biomarkers such as hsCRP (*p* = 0.57), LCR (*p* = 0.78), or CAR (*p* = 0.78) (Table 2).

Obesity is a prevalent driver of abnormal metabolic and cardiovascular risk factors and the TyG index is a marker of insulin resistance. Two pathophysiological mechanisms that may explain why the TyG index is a good marker of insulin resistance have been proposed. First, the increased flow of fatty acids and triglycerides may cause insulin resistance in the muscle, liver, and adipose tissue. Second, glucose lipotoxicity may also be related to the development of insulin resistance [13]. TyG index values have been found to be associated with changes in total fat mass even after adjusting for age, sex, smoking status, baseline cholesterol, or systolic blood pressure, among other parameters [18].

Not all patients with obesity as determined using BMI have the alterations in cardiovascular risk factors that are expected from excess body fat. Therefore, other parameters should also be used in order to more precisely define their risk. WC, for example, is a useful index of abdominal adiposity. For patients in every BMI category, an elevated WC was predictive of an increased accumulation of visceral adipose tissue (VAT) and mortality. Kissebah and Björntorp first reported that in all BMI categories, a greater proportion of abdominal fat was predictive of insulin resistance and hypertriglyceridemia. This finding is in line with our results and provides the first evidence of the limitations of BMI in a patient’s health risk assessment [19,20]. Current guidelines recommend sex-specific WC values (>102 cm in men and >88 cm in women) to define abdominal obesity [21]. However, these cutoff values are not BMI-specific. A WC of 103 cm does not describe the same adiposity phenotype in a man with a BMI of 29 Kg/m^2^ as in another with a BMI of 35 Kg/m^2^. In this case, the first man has central obesity with excess VAT and should be considered a high-risk patient. In our sample, 54.5% had class I obesity and 44.8% had abdominal obesity; in contrast, 41.42% had class II obesity and only 37.3% had abdominal obesity. Neither BMI nor WC should be used on their own to define individuals with obesity [22]. In a metanalysis of 11 prospective cohort studies, Cerhan et al. observed that WC was positively associated with mortality within every BMI category [23]. Similarly, de Hollander et al. reported that age-adjusted and smoking-adjusted mortality was greater for those with an elevated WC in every BMI category [24]. It appears that the amount of intra-abdominal adipose tissue—not subcutaneous adipose tissue—critically correlates with the atherogenic and diabetogenic metabolic abnormalities observed among individuals with obesity, an assertion supported by the results of this study [25]. These findings provide a plausible mechanism by which lower BMI values in an individual with a high-risk WC would entail increased adverse health risk, especially for older adults such as those in our study, whose mean age was 64 years [26].

In light of the foregoing, questions may arise about the homogeneity of the group classified as “high-risk patients” in this work. Is the risk the same in all patients? Can it be assumed that adipose tissue is deposited in the same way in all patients? Likewise, do all obesity phenotypes entail the same risk from a metabolic and inflammatory point of view? The need to classify these “*high-risk obesity*” into phenotypes and determine their relationships seems clear. In this work, the two clusters created from the high-risk obesity group exemplify two distinct high-risk phenotypes (Graphical Abstract). Patients in cluster 2 were older and had more CVD, heart failure, sarcopenia, and disability. Interestingly, they also had a greater inflammatory burden. This may be because VAT is a metabolically active organ that, on the one hand, has a particular metabolism, including an active lipolysis that contributes to impairing metabolic functions and, on the other hand, becomes infiltrated with inflammatory M2 macrophages and is a source of low-grade chronic inflammation through the production of adipokines such as interleukin-6 (IL-6) and tumor necrosis factor-alpha [27,28]. Increased circulating IL-6 levels are associated with coronary artery disease, promote vascular inflammation, and increase circulating CRP levels [29]. CRP provides prognostic information on cardiovascular risk that is comparable to blood pressure or cholesterol. Values < 1, 1 to 3, and >3 mg/L indicate lower, average, or higher relative cardiovascular risk, respectively [30]. Inflammation may lead to low levels of serum albumin through a decrease in its synthesis and an increase in its degradation. Albumin has several anti-atherogenic properties, including antioxidant activities as well as the inhibition of platelet activation and aggregation through the modulation of arachidonic acid metabolism. New models of risk based on inflammation, which include parameters such as CAR, platelets, the neutrophil to lymphocyte ratio, and LCR, better reflect inflammatory status. This was first reported in diseases such as COVID-19, cancer, CVD, or diabetic nephropathy [31,32]. CAR reflects both oxidative stress and antioxidant status and could be used as a surrogate inflammatory marker. CAR has also been observed in CVD. Kelesoglu et al. reported that increased CAR may predict poor coronary collateral circulation in patients with coronary heart disease [33]. The LCR ratio is of great potential interest as a prognostic marker because it reflects both the systemic inflammatory response and the immune response [32]. A common feature of these conditions is low-grade chronic inflammation, which is also a characteristic of obesity. To the best of our knowledge, this is the first study to link the LCR ratio to the presence of inflammation and established CVD in high-risk obese patients. The increased CAR and LCR reported in our study are findings consistent with relevant knowledge in the literature.

The importance of the NLRP3 inflammasome in immunity and human diseases has been well documented [34]. The assembling of NLRP3 inflammasome leads to the activation of caspase-1-mediated inflammatory responses, including cleavage secretion of the proinflammatory cytokines-IL-18 and IL-1β [35]. Recent findings suggest a different role of the inflammasome NLRP3 among the various obesity classes. Thus, in a very elegant study, Antonioli et al. explored the inflammasome NLRP3 activation in patients with morbid obesity and T2D compared with patients with morbid obesity and normal glucose tolerance after treatment by Roux-en-Y gastric bypass [36]. Interestingly, plasma caspase-1 concentrations normalized in both groups, whereas plasma IL-1β levels normalized only in patients with morbid obesity and normal glucose tolerance suggesting the persistence of a systemic inflammatory condition in people with T2D. Unfortunately, in our study, we did not measure the regulation of NLRP3 inflammasome. However, this hypothesis could explain the observed diversity of inflammation indices between cluster 1 vs. cluster 2 within the patients of the 4 group.

In our study, two subprofiles of subjects with “*high-risk obesity*” were identified. Older subjects with obesity require measures that target sarcopenia, disability, psychological health, and significant comorbidities to prevent further deterioration of their health. Longitudinal studies should be performed to identify potential risk factors of subjects who progress from cluster 1 to cluster 2.

The limitations of our study include the fact that it is an observational study, which implies the existence of several variables that are not controlled a priori. As it is a cross-sectional study, the progress of the patients included in the study or the progress of those within the clusters are not known. We must be cautious when interpreting our results. Some of the observed groups have a very low number of patients, which does not allow generalizations to be made based on the results obtained in that group. This study also has several strengths. Two different methods of classifying patients were used: one based on tools used in routine clinical practice and another more complex classification that was not supervised a priori, which could detect different phenotypes of patients within group 4, the “highest risk” obesity category.

## 6. Conclusions

Obesity is associated with an increased disease burden consisting of a variety of comorbidities and may be an important target for multimorbidity prevention. It is unclear whether these diseases are distributed among all individuals with obesity or are clustered in smaller groups of individuals with obesity-related multimorbidity. Therefore, stratifying obesity into risk-based classifications would allow us to identify individuals with “*high-risk obesity*”, better and earlier, or identify those whose excess adiposity entails an increased risk to their health. WC is a body measurement that encompasses both subcutaneous and visceral adipose tissues, some visceral organs or skeletal muscle, and could be used to help stratify individuals with obesity. Along with BMI and WC, the role of inflammation in the development of “*high-risk obesity*” must also be taken into account. The use of new biomarkers based on insulin resistance, such as the TyG index, or the role of inflammation and immunomodulation in the pathogenesis of obesity-related diseases, such as CAR or LCR ratio, are necessary for classifying individuals with obesity and identifying those with high-risk phenotypes.

## Figures and Tables

**Figure 1 jcm-11-04644-f001:**
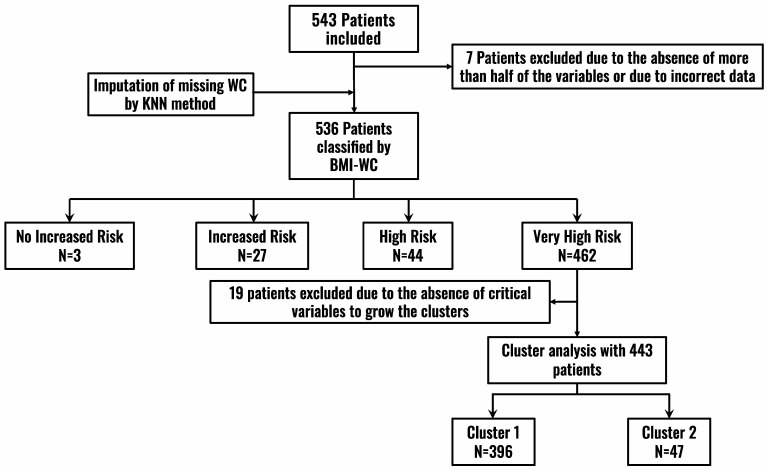
Patient inclusion flowchart. BMI: Body Mass Index; KNN: k-nearest neighbors (KNN) algorithm; WC: Waist Circumference.

**Figure 2 jcm-11-04644-f002:**
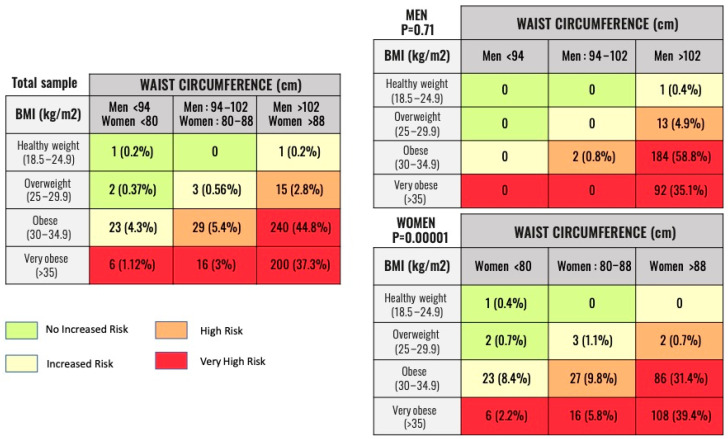
Number and percentage of patients belonging to the categories of Obesity Classification by BMI/WC relationship. Legend: Classification of patients according to metabolic risk. The NICE recommendations have been used based on the value of BMI and WC. The results of the total sample and by gender are shown. BMI: Body Mass Index. WC: Waist Circumference.

**Figure 3 jcm-11-04644-f003:**
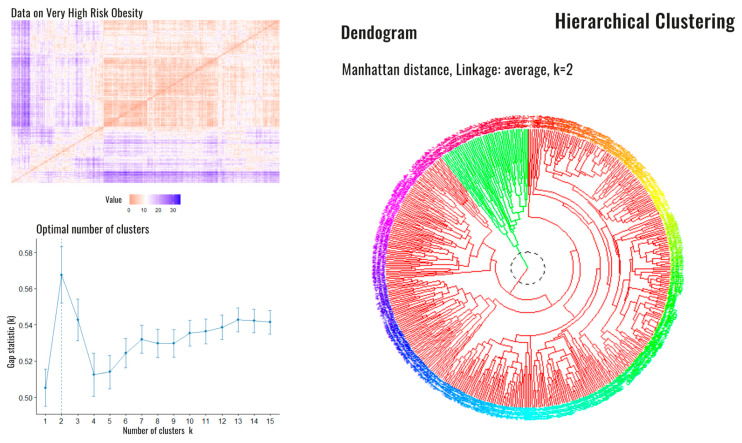
Dissimilarity matrix, optimal number of clusters and dendrogram corresponding to high-risk patients. Legend: Dissimilarity matrix (**upper left**), the optimal number of estimated Clusters (**down left**). On the right side, the dendrogram is shown with the presence of the two clusters: cluster 1 (red) and cluster 2 (green).

**Table 1 jcm-11-04644-t001:** Qualitative Variables by Groups.

Variable (Absolute Number (Percentage))	Group 1 *n* = 3	Group 2 *n* = 27	Group 3 *n* = 44	Group 4 *n* = 462	*p*
Sex (women)	3 (100)	26 (96.3)	29 (65.9)	216 (46.7)	** *0.0001* **
Smoker	0	0	9 (20.4)	72 (15.6)	0.09
Education level					0.14
Illiterate	0	1 (3.7)	2 (4.8)	33 (7.3)	
Primary	0	14 (5.8)	27 (64.3)	190 (41.9)	
Secondary	2 (66.7)	7 (25.9)	10 (23.8)	170 (37.5)	
University	1 (33.3)	5 (18.5)	3 (7.14)	60 (13.2)	
Employed	1 (33.3)	10 (37.04)	16 (36.4)	177 (38.4)	0.99
Origin (Urban)	2 (66.7)	18 (66.7)	30 (68.2)	349 (75.5)	0.5
Physical activity	2 (66.7)	8 (29.6)	9 (20.4)	157 (33.9)	0.17
HBP	0	13 (48.1)	36 (81.8)	331 (71.6)	** *0.0008* **
T2DM	0	5 (18.5)	17 (38.6)	199 (43.1)	** *0.03* **
Dyslipidemia	0	12 (44.4)	29 (65.9)	313 (43.1)	** *0.007* **
Hyperuricemia	0	1 (3.7)	1 (2.27)	102 (22.2)	** *0.001* **
HFpEF	0	3 (11.1)	6 (13.6)	69 (14.9)	0.85
HFrEF	0	0	1 (2.3)	19 (4.1)	0.85
CAD	0	2 (7.4)	6 (13.6)	42 (9.1)	0.7
Stroke	0	0	5 (11.3)	35 (7.6)	0.3
Gastroesophageal reflux disease	1 (33.3)	8 (29.6)	7 (15.9)	67 (14.5)	0.15
COPD/Asthma	0	2 (7.4)	5 (11.4)	48 (10.4)	0.88
Cancer	1 (33.3)	0	0	13 (2.8)	** *0.004* **
Arthrosis	1 (33.3)	7 (25.9)	21 (47.7)	141 (30.5)	0.11
Depression	0	6 (22.2)	11 (25)	98 (21.3)	0.76
Disability					
Moderate	0	2 (7.4)	11 (25)	93 (20.3)	0.35
Severe	0	1 (3.7)	5 (11.4)	42 (9.15)	0.35
Previous treatment					
Glucocorticoids	0	2 (7.4)	3 (6.8)	28 (6.1)	0.9
Metformin	0	3 (11.1)	14 (31.8)	156 (33.9)	0.05
Sulfonylureas	0	1 (3.7)	3 (6.8)	9 (1.9)	0.2
DPP-4 inhibitors	0	1 (3.7)	3 (6.8)	38 (8.3)	0.78
GLP-1 RA	0	1 (3.7)	5 (11.4)	78 (16.9)	0.19
SGLT2 inhibitors	0	3 (11.1)	6 (13.6)	79 (17.2)	0.6
Insulin	0	2 (7.4)	10 (22.7)	54 (11.7)	0.13
Statins	0	9 (33.3)	26 (59.1)	259 (56.3)	** *0.02* **
IBP	1 (33.3)	13 (48.15)	31 (70.4)	231 (50.2)	0.06
Antihypertensives	0	14 (51.8)	33 (75)	325 (70.8)	** *0.007* **
NSAIDs	0	6 (22.2)	11 (25)	65 (14.2)	0.15
Antidepressants	0	6 (22.2)	15 (34.1)	104 (22.6)	0.27

Legend: CAD: coronary artery disease; COPD: chronic obstructive pulmonary disease; DPP-4 inhibitors: dipeptidyl-dipeptidase 4 inhibitors; GLP-1 RA: glucagon like peptide-1 receptor agonist; HBP: high blood pressure; HFpEF: heart failure with preserved ejection fraction; HFrEF: heart failure with reduced ejection fraction; IBP: protons bomb inhibitors; NSAIDs: nonsteroidal anti-inflammatory drugs; SGLT2 inhibitors: sodium-glucose cotransporter 2 inhibitors; T2DM: type 2 diabetes mellitus. Data are expressed by absolute number and percentage.

**Table 2 jcm-11-04644-t002:** Quantitative Variables by Groups.

Variable (Median/Interquartile Range)	Group 1 *n*= 3	Group 2 *n* = 27	Group 3 *n* = 44	Group 4 *n* = 462	*p*
Age (years)	62 (26)	59 (22.5)	65 (22.2)	62 (22)	0.17
Weight (Kg)	64 (21)	78 (8.8)	81 (9.6)	97 (21.9)	** *0.0000* **
BMI (Kg/m^2^)	26.6 (7.2)	31.9 (2.7)	31.2 (3.7)	34.7 (6.9)	** *0.0000* **
WC (cm)	81 (10.5)	90 (5)	100 (9.2)	112 (13.5)	** *0.0000* **
Charlson	0 (3)	0 (1)	1 (2)	1 (3)	** *0.009* **
FPG (mg/dL)	89 (7.5)	97 (18.5)	101 (28)	104 (32)	** *0.04* **
HbA1c (%)	5.3 (0.5)	5.7 (0.87)	6 (0.8)	5.9 (1.3)	0.1
eGFR (ml/min/1.73 m^2^)	66.9 (41.9)	87.6 (24.3)	85.6 (33.6)	84.1 (36)	0.28
Uric acid (mg/dL)	2.8 (0.15)	4.65 (1.85)	5.18 (2.01)	5.8 (2.5)	** *0.000* **
hsCRP (mg/dL)	1.6 (1.3)	3 (7.9)	2 (3)	3 (5.5)	0.57
LDL-c (mg/dL)	96.4 (45.5)	107 (44)	109 (61)	97 (52)	0.85
HDL-c (mg/dL)	57 (4.5)	53 (15)	49.5 (22)	46 (16)	** *0.0002* **
Triglycerides (mg/dL)	113 (74.5)	102 (60)	119.5 (72)	136 (86)	** *0.009* **
TyG index	9.1 (0.7)	9.3 (0.7)	9.5 (0.9)	9.6 (0.7)	** *0.008* **
AST (U/L)	46 (21)	23 (19.5)	20 (15.7)	22 (17)	0.63
ALT (U/L)	30 (15.5)	24 (17.5)	20 (12)	21 (12.7)	0.85
GGT (U/L)	72 (47)	24 (34)	28.5 (48.5)	32 (31)	0.07
ALP (U/L)	83 (27.5)	75.5 (38.7)	83 (52)	79 (35)	0.48
Hemoglobin (g/dL)	13.8 (0.1)	13.6 (1.45)	13.6 (2.4)	14 (2.3)	** *0.016* **
Leukocytes (×10^9^/L)	6.9 (1.15)	7.5 (2.5)	6.34 (3.1)	7.4 (2.8)	0.06
Lymphocytes (×10^9^/L)	2.5 (0.6)	2.31 (0.93)	1.96 (1.18)	2.13 (1.1)	0.49
Platelets (×10^9^/L)	256 (110.5)	224 (132)	228 (75.2)	237 (97)	0.66
Albumin (g/dL)	4.2 (0.3)	4.1 (0.45)	4.2 (0.5)	4.3 (0.5)	0.26
UACR (mg/g)	7.5 (4.5)	11.2 (10.7)	11.1 (20.6)	9.4 (19.4)	0.9
Drugs number	2 (1)	4.5 (4.7)	8 (4)	7 (6.25)	** *0.004* **

Legend: ALP: alkaline phosphatase; ALT: Alanine transaminase; AST: Aspartate transaminase; BMI: body mass index; eGFR: estimated glomerular filtration rate; FPG: Fasting plasma glucose; GGT: gamma-glutamyl transferase; HbA1c: glycated hemoglobin; HDL-C: high density lipoprotein cholesterol; hsCRP: high-sensitivity C-reactive protein; LDL-C: low density lipoprotein cholesterol; TyG index: triglyceride-glucose index; UACR: Urinary albumin-to-creatinine ratio; WC: waist circumference. Data is expressed as median and interquartile range since non normality of the data.

**Table 3 jcm-11-04644-t003:** Cluster Analysis in high-risks patients.

Variable	Cluster 1 *n* = 396	Cluster 2 *n* = 47	*p*
Origin (Urban)	299 (74.9)	36 (76.6)	0.8
Age (years)	61 (21)	77 (16)	** *0.00* **
Sex (women)	185 (46.4)	27 (57.4)	0.15
SARC-F (>4)	72 (16)	37 (78.7)	** *0.00* **
HBP	275 (68.9)	42 (89.4)	** *0.003* **
T2DM	169 (42.4)	25 (53.2)	0.15
Dyslipidemia	265 (66.4)	34 (72.3)	0.41
Hyperuricemia	75 (18)	19 (40.4)	** *0.0006* **
HF	57 (14.3)	27 (57.4)	** *0.0001* **
CAD	27 (6.7)	10 (21.3)	** *0.0006* **
Stroke	22 (5.5)	12 (25.5)	** *0.00* **
GERD	53 (13.3)	10 (21.3)	0.13
COPD/Asthma	43 (10.8)	5 (10.6)	0.97
Cancer	8 (2)	5 (10.6)	** *0.0009* **
Arthrosis	109 (27.3)	29 (61.7)	** *0.0001* **
Depression	78 (19.5)	17 (36.2)	** *0.008* **
Disability			
Moderate	77 (19.3)	12 (25.5)	** *0.000* **
Severe	12 (3.01)	30 (63.8)	** *0.000* **
Comorbidities Charlson Index			
Mild (0–1)	316 (79.2)	4 (8.5)	** *0.000* **
Moderate (2)	64 (16)	18 (38.3)	** *0.000* **
Severe (≥3)	19 (4.8)	25 (53.2)	** *0.000* **
BMI (kg/m^2^)	34.7 (6.6)	34.5 (8.89)	0.46
WC (cm)	112 (13.9)	116.3 (15.7)	** *0.02* **
FPG (mg/dL)	104 (32)	106.5 (37)	0.79
eGFR (mil/min/1.73 m^2^)	86.1 (30.4)	52.1 (42.4)	** *0.000* **
Uric acid (mg/dL)	5.8 (2.2)	7.4 (3.3)	** *0.00* **
hsCRP (mg/dL)	2.1 (4.5)	5.8 (4.1)	** *0.008* **
HDL-Chol (mg/dL)	46 (15)	48 (16)	0.63
LDL-Chol (mg/dL)	97 (51)	93 (67.5)	0.9
Triglycerides (mg/dL)	136 (87)	134.3 (72)	0.95
HbA1c (%)	5.9 (1.2)	6.3 (2)	** *0.05* **
AST (U/L)	23 (17)	15.5 (12.2)	** *0.001* **
ALT (U/L)	21 (12)	18 (7.5)	** *0.01* **
GGT (U/L)	31 (31)	35 (36.5)	0.11
ALP (U/L)	79 (35)	79.5 (37.5)	0.3
Hemoglobin (g/dL)	14.1 (2.3)	13 (2.2)	** *0.000* **
Leukocytes (×10^9^/L)	7.4 (2.7)	8.3 (3.2)	** *0.007* **
Lymphocytes (×10^9^/L)	2.2 (1)	1.9 (1.7)	0.33
Platelets (×10^9^/L)	237 (95)	257 (115)	0.97
Albumin (mg/dL)	4.3 (0.5)	3.9 (0.7)	** *0.000* **
UACR (mg/g)	9 (17.8)	15 (38.2)	0.17
TyG index	4.1 (0.3)	4.1 (0.3)	0.91
C-reactive Protein/Albumin ratio (CAR)	0.5 (1.1)	1.3 (3.4)	** *0.01* **
Lymphocyte to CRP ratio (LCR)	0.925 (1.86)	0.52 (0.9)	** *0.002* **

Legend: ALP: alkaline phosphatase; ALT: Alanine transaminase; AST: Aspartate transaminase; BMI: body mass index; CAD: coronary artery disease; COPD: chronic obstructive pulmonary disease; eGFR: estimated glomerular filtration rate; FPG: fasting plasma glucose; FPG: Fasting plasma glucose; GERD: gastroesophageal reflux disease; GGT: Gamma-glutamyl transferase; HBP: high blood pressure; HbA1c: glycated hemoglobin; HDL: high density lipoprotein; hsCRP: high-sensitivity C-reactive protein; HF: heart failure; LDL: low density lipoprotein; T2DM: type 2 diabetes mellitus; UACR: Urinary albumin-to-creatinine ratio; SARC-F: Strength, Assistance in walking, Rise from a chair, Climb stairs, and Falls questionnaire; WC: waist circumference.

## Data Availability

The datasets generated during and/or analyzed during the current study are available from the corresponding author on reasonable request.

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
