# Peer review of "High-Risk Obesity Phenotypes: Target for Multimorbidity Prevention at the ROFEMI Study"

_jcm, 2022, doi:10.3390/jcm11164644_

Round 1

Reviewer 1 Report

Although the paper offers some interesting aspects, including the fact that BMI alone is not enough to define the severity of the obesity, and brings to light the presence of a particular cluster of obese people with the worst profile, there are some flaws that do not make the paper suitable for publication. 

-       All the statistical analysis inherent to the results and to table 1 is to be completely revised. It is not possible to include a group of only 3 patients, such as group 1 (moreover arguing that it is at greater risk of cancer), the Kruskal Wallis test and the ANOVA on a sample of such sample size loses any statistical value. Rather, a statistical analysis could be performed between group 4 versus all the rest of the patients, and subsequently show the statistical analysis of only the patients within group 4. -       Since there is a diversity of inflammation indices between the 2 clusters within the patients of the 4 group (cluster 1 vs cluster 2), but not between the very high-risk patients and the low-risk patients, it would be useful to explain the interpretation of these results more fully in discussion. A paper was recently published showing the role of the inflammasome NLRP3 among the various obesity classes, (see PMID: 32020775) and it may be useful to briefly mention this inflammatory cascade when discussing the discrepancies with other inflammation indices such as CRP.

-       There are many typos that need to be correct throughout the text. I give some examples. Page 2, row 51 “cincumference” and “obesities”; Hyperglucemia (Graphical Abstract) and so on.

Author Response

REVIEWER 1:

We appreciate your thorough review and constructive suggestions to improve the quality of our paper. Accordingly, we made the changes detailed below and highlighted in yellow in the manuscript document.

  • Question 1. All the statistical analysis inherent to the results and to table 1 is to be completely revised. It is not possible to include a group of only 3 patients, such as group 1 (moreover arguing that it is at greater risk of cancer), the Kruskal Wallis test and the ANOVA on a sample of such sample size loses any statistical value. Rather, a statistical analysis could be performed between group 4 versus all the rest of the patients, and subsequently show the statistical analysis of only the patients within group 4.

Answer: Thanks for your comment. Tables 1 and 2 reflect the actual distribution of patients seen in our clinics according to the classification established in the NICE guidelines. We fully agree with the reviewer that some groups contain a very low number of patients, but grouping these subgroups means losing the categories defined and accepted by the NICE guidelines (https://www.guidelines.co.uk/public-health/nice-obesity-guideline/252547.article); Following your indication, we have grouped data and generated some new information according to your request (see tables below). As you can observe, there are variations in the significance of some of the variables. As we mention before, the interpretation of these results would be assuming a classification of the patients that has been made by cut-off points (BMI/WC) not previously validated. For this reason, and despite the small number of patients, and in order not to lose the information of each group validated by the NICE guidelines, we have maintained the initial tables. If the reviewer thinks it necessary, we can include the new information with only two groups as supplementary material. In addition, we have also added the following text in the limitations: “We must be cautious when interpreting our results. Some of the observed groups have a very low number of patients, which does not allow generalizations to be made based on the results obtained in that group”.

Table 1. Qualitative Variables by Groups

Variable

(Absolute number (percentage))

Group 1

n=74

Group 2

n=462

p

Sex (women)

58 (78.4)

216 (46.7)

0.000

Smoker

9 (12.2)

72 (15.6)

0.44

Education level

Illiterate

Primary

Secondary

 University

(n=72)

3 (4.2)

41 (56.9)

19 (26.4)

9 (12.5)

(n=453)

33 (7.3)

190 (41.9)

170 (37.5)

60 (13.2)

0.10

Employed

27 (36.5)

177 (38.4)

0.7

Origin (Urban)

50 (67.6)

349 (75.5)

0.14

Physical activity

19 (25.7)

157 (33.9)

0.15

HBP

49 (66.2)

331 (71.6)

0.33

T2DM

      22 (29.7)

199 (43.1)

0.03

Dyslipidemia

41 (55.4)

313 (43.1)

0.03

Hyperuricemia

2 (2.7)

102 (22.2)

0.0001

HFpEF

9 (12.2)

69 (14.9)

0.39

HFrEF

1 (1.3)

19 (4.1)

0.39

CAD

 8 (10.8)

42 (9.1)

0.63

Stroke

5 (6.8)

35 (7.6)

0.8

Gastroesophageal reflux disease

16 (21.6)

67 (14.5)

0.11

COPD/Asthma

7 (9.5)

48 (10.4)

0.8

Cancer

1 (1.35)

13 (2.8)

0.46

Arthrosis

29 (39.2)

141 (30.5)

0.14

Depression

17 (22.9)

98 (21.3)

0.73

Disability

Moderate

Severe

13 (17.6)

6 (8.1)

93 (20.3)

42 (9.15)

0.8

0.8

Previous treatment

Glucocorticoids

Metformin

Sulfonylureas

DPP-4 inhibitors

GLP-1 RA

SGLT2 inhibitors

Insulin

Statins

IBP

Antihypertensives

NSAIDs

Antidepressants

5 (6.8)

17 (22.9)

4 (5.4)

4 (5.4)

6 (8.1)

9 (12.2)

12 (16.2)

35 (47.3)

45 (60.8)

47 (63.5)

17 (22.9)

21 (28.4)

28 (6.1)

156 (33.9)

9 (1.9)

38 (8.3)

78 (16.9)

79 (17.2)

54 (11.7)

259 (56.3)

231 (50.2)

325 (70.8)

65 (14.2)

104 (22.6)

0.82

0.06

0.07

0.39

0.05

0.27

0.27

0.14

0.09

0.2

0.05

0.28

Legend: CAD: coronary artery disease; COPD: chronic obstructive pulmonary disease; DPP-4 inhibitors: dipeptidyl-dipeptidase 4 inhibitors; GLP-1 RA: glucagon like peptide-1 receptor agonist; HBP: high blood pressure; HFpEF: heart failure with preserved ejection fraction; HFrEF: heart failure with reduced ejection fraction; IBP: protons bomb inhibitors; NSAIDs: nonsteroidal anti-inflammatory drugs; SGLT2 inhibitors: sodium-glucose cotransporter 2 inhibitors; T2DM: type 2 diabetes mellitus. Data are expressed by absolute number and percentage.

Table 2. Quantitative Variables by Groups

Variable

(median/interquartile range)

Group 1

n= 74

Group 2

n=462

p

Age (years)

61 (27)

62 (22)

0.87

Weight (Kg)

79.9 (11)

97 (22)

0.0000

BMI (Kg/m2)

31.3 (3.7)

34.7 (6.9)

0.0000

WC (cm)

97.5 (10)

112 (13.5)

0.0000

Charlson

1 (2)

1 (3)

0.002

FPG (mg/dL)

100 (26)

104 (32)

0.04

HbA1c (%)

5.8 (0.7)

5.9 (1.3)

0.32

eGFR (ml/min/1.73m2)

86.3 (27.3)

84.1 (36)

0.29

Uric acid (mg/dL)

5 (1.8)

5.8 (2.5)

0.0000

hsCRP

(mg/dL)

2 (4)

3 (5.5)

0.36

LDL-c

(mg/dL)

107 (60)

97 (52)

0.57

HDL-c

(mg/dL)

52 (17)

46 (16)

0.0001

Triglycerides (mg/dL)

114 (72)

136 (86)

0.002

TyG index

8.6 (0.8)

9.6 (0.7)

0.003

AST (U/L)

22 (17)

22 (17)

0.6

ALT (U/L)

21.5 (20.5)

21 (12.7)

0.56

GGT (U/L)

25 (47.5)

32 (31)

0.07

ALP (U/L)

81.5 (42)

79 (35)

0.53

Hemoglobin (g/dL)

13.7 (2)

14 (2.3)

0.001

Leukocytes (x109/L)

6.8 (2.97)

7.4 (2.8)

0.01

Lymphocytes (x109/L)

2.12 (1.2)

2.13 (1.1)

0.53

Platelets (x109/L)

228 (82)

237 (97)

0.33

Albumin (g/dL)

4.2 (0.5)

4.3 (0.5)

0.06

UACR (mg/g)

11.1 (17.1)

9.4 (19.4)

0.8

Drugs number

6 (6)

7 (6.25)

0.79

Legend: ALP: alkaline phosphatase; ALT: Alanine transaminase; AST: Aspartate transaminase; BMI: body mass index; eGFR: estimated glomerular filtration rate; FPG: Fasting plasma glucose; GGT: gamma-glutamyl transferase; HbA1c: glycated hemoglobin; HDL-C: high density lipoprotein cholesterol; hsCRP: high-sensitivity C-reactive protein; LDL-C: low density lipoprotein cholesterol; TyG index: triglyceride-glucose index; UACR: Urinary albumin-to-creatinine ratio; WC: waist circumference. Data is expressed as median and interquartile range since non normality of the data.

Reviewer 2 Report

Interesting paper about the study that aimed to identify different profiles of patients with high-risk obesity from a cluster analysis, which allowed the authors to identify two profiles of individuals with high-risk obesity. Methodology according to the intended objective. Discussion and conclusions appropriate to the results obtained.

What is the main question addressed by the research?

- The identification of two profiles of individuals with high risk obesity.

Is it relevant and interesting?

- I think yes.

How original is the topic?

- It was new to me.

What does it add to the subject area compared to other published materials?

- Improved identification of high-risk obese individuals.

Is the paper well written?

- I think yes.

Is the text clear and easy to read?

- It was easy to read.

Are the conclusions consistent with the evidence and arguments presented?

- Yes, the conclusions are based on the results obtained.

Do they address the main question posed?

- I think yes.

Author Response

REVIEWER 2:

We appreciate your thorough review and constructive suggestions to improve the quality of our paper. Overall, we´d like to thank the reviewer for the relevant and pertinent comments, which show its mastery in the topic. Thanks to the changes made after these comments, the article has improved its quality, and we are grateful about that.

Reviewer 3 Report

The manuscript “High-risk obesity phenotypes: target for multimorbidity prevention at the ROFEMI study” describes a study which investigated characteristics of High-Risk Obesity Phenotype by using a cluster analysis. The study is straight-forward, the results are presented well and are interesting. There are a few minor comments.

1.       For the Tables, the far right column has the p value, but because the tables are extensive, it would be helpful to include another indication of significance (ex. Bold font, italicized font, asterisk, etc).

2.       For the Tables, it is unclear what is number represents and what is in the parentheses. For Table 1, is it Number (Percentage)?, Table 2, Value (??)? etc. Please add to figure legends. Also, is Table 3 supposed to be before Tables 1 & 2 in the text?

Author Response

REVIEWER 3:

We appreciate your thorough review and constructive suggestions to improve the quality of our paper. Accordingly, we made the changes detailed below and highlighted in yellow in the manuscript document.

The manuscript “High-risk obesity phenotypes: target for multimorbidity prevention at the ROFEMI study” describes a study which investigated characteristics of High-Risk Obesity Phenotype by using a cluster analysis. The study is straight-forward, the results are presented well and are interesting. There are a few minor comments.

  • Question 1: For the Tables, the far-right column has the p value, but because the tables are extensive, it would be helpful to include another indication of significance (ex. Bold font, italicized font, asterisk, etc).

Answer: Thank you for the advice. We have modified the tables by indicating in bold and italics the cells with statistical significance in the column corresponding to the value of “p”.

  • Question 2: For the Tables, it is unclear what is number represents and what is in the parentheses. For Table 1, is it Number (Percentage)?, Table 2, Value (??)? etc. Please add to figure legends. Also, is Table 3 supposed to be before Tables 1 & 2 in the text?

Answer:

  • Thank you for the advice. We have referred in the legend and in the corresponding table the meaning of the value between parentheses.
  • We have added a legend in the figure 1, the other figures already included an explanatory legend.
  • Yes, the reviewed is right. In the original submitted manuscript the table 3, corresponding to the two clusters, is before to tables 1 and 2.

      Overall, we´d like to thank the reviewer for the relevant and pertinent comments, which show its mastery in the topic. Thanks to the changes made after these comments, the article has improved its quality, and we are grateful about that.

Round 2

Reviewer 1 Report

In light of the changes made by the authors, I believe that this paper can be suitable for publication in this journal.